# Epidemiology and risk factors of coronary artery aneurysm in Taiwan: a population based case control study

Chein-Tang Fang,[1] Yi-Ping Fang,[1] Yaw-Bin Huang,[1,2] Chen-Chun Kuo,[2] Chung-Yu Chen[1,2]

► Prepublication history and additional material are available. To view these files please visit the journal online (http://dx.doi.org/10.1136/bmjopen-2016-014424)

[1]School of Pharmacy, Master Program in Clinical Pharmacy, Kaohsiung Medical University, Kaohsiung, Taiwan
[2]Department of Pharmacy, Kaohsiung Medical University Hospital, Kaohsiung, Taiwan

**Correspondence to**
Chung-Yu Chen; jk2975525@hotmail.com

## ABSTRACT

**Objectives** Coronary artery aneurysm (CAA) is usually an asymptomatic and rare disease. There are limited epidemiological data for CAA in Asian populations and in the rest of the world.

**Design** A retrospective case control study.

**Setting** A population based, database study from Taiwan's National Health Insurance Research Database, between 2005 and 2011.

**Participants** CAA patients identified using International Classification of Diseases, ninth revision, clinical modification (ICD-9-CM) code 414.11 with CAA examinations.

**Outcome measures** The incidence rate and mortality rate of CAA were calculated. We also matched patients with non-CAA patients according to age, gender and index year at a 1:10 ratio to explore the risk factors for CAA using conditional logistic regression.

**Result** A total of 1397 CAA patients were identified between 2005 and 2011; 41.9% were paediatric patients and 58.1% were adults. The incidence rate and mortality rate of CAA in Taiwan were 0.87 and 0.05 per $10^5$ person-years, respectively. The adjusted odds ratios (aOR) for coronary atherosclerosis, hypertension, dyslipidaemia and diabetes were 7.97, 2.09, 2.48 and 1.51, respectively. Of note, aortic dissection (aOR 6.76), aortic aneurysm (aOR 5.82) and systemic lupus erythematosus (aOR 4.09) were found to be significantly associated with CAA.

**Conclusion** In Taiwan, CAA patients were distributed across both paediatric and adult populations. Apart from cardiovascular risk factors, aortic diseases and systemic lupus erythematosus need to be investigated further in CAA patients.

## INTRODUCTION

Coronary artery aneurysm (CAA) is a rare finding in patients referred for coronary angiography. It is characterised by abnormal dilatation of the vessel lumen, defined as a lumen with a diameter >1.5-fold the diameter of the lumens of the normal vessels adjacent to it or the largest coronary vessel lumen.[1] Previous case series on angiography have reported a wide prevalence range, from 0.2% to 6.0%, across the world.[1–4] In Taiwan, Wang *et al* reported 25 patients with CAA (0.25%) in

### Strengths and limitations of this study

► In this nationwide epidemiology study involving retrospective analysis of a claims database, we evaluated the epidemiology and risk factors for coronary artery aneurysm in an Asian country.
► The major limitation of the study was no laboratory data on coronary angiography and lifestyle habits of the patients.
► As a retrospective observational analysis, this study only provides associative information from a case control study.

a patient cohort of 10 120 patients.[5] Furthermore, one case series indicated that there were 24 patients (2.6%) with aneurysmal dilatation among 924 acute myocardial infarction patients undergoing percutaneous coronary intervention from 1993 to 2001.[6]

Although CAA commonly coexists with atherosclerosis,[1 7] various potential causes, such as congenital, inflammatory and connective tissue disorders, and other factors, have been noted in the literature.[8 9] Among these risk factors, Kawasaki disease is well recognised as the main cause in children; it occurs predominantly in Asian countries.[10 11] The pathogenesis of CAA is postulated to be the degradation of the extracellular matrix of the media by matrix metalloproteinases and is considered to be similar to that of other aneurysms in larger vessels.[12 13] Instead of presenting as a benign entity, CAA shows the clinical manifestations of coronary artery diseases, such as angina and acute coronary syndrome.[14–16]

Despite the fact that Kawasaki disease is a significant risk factor for the development of CAA, especially in children, information regarding CAA and its related risk factors in the adult population is limited in Asian countries. Accordingly, our aim was to investigate the epidemiology and risk factors for CAA in a Taiwanese population.

## METHODS

### Data sources

We obtained a specific dataset of all CAA patients from Taiwan's National Health Insurance Research Database (NHIRD). All medical records comprising outpatient care, inpatient care, emergency care and prescriptions from 2005 to 2011 were used for analysis in the epidemiological study.

For the case control study, we selected controls from the Longitudinal Health Insurance Database (LHID2005), which contained 1 million samples that were randomly selected from the NHIRD and were representative of 23 million people in Taiwan.

The study was approved by the institutional review board of Kaohsiung Medical University Hospital (KMUH-IRB-EXEMPT-20130199). NHIRD data were de-identified by scrambling the identification codes of both patients and medical facilities, and were released to the public for research purposes. Therefore, current NHIRD and hospital regulations and guidelines did not mandate informed consent in this retrospective case control study because of the anonymous nature of the database. All procedures performed were in accordance with the ethical standards of the institutional research committee and the directives of the Declaration of Helsinki.

### Study population

To identify patients diagnosed with CAA, we used the International Classification of Diseases, ninth revision, clinical modification (ICD-9-CM) code 414.11. Patients who had two outpatient diagnoses or one inpatient diagnosis of CAA between 1 January 2005 and 31 December 2011 were identified. The index date was defined as the date of first diagnosis of CAA. Patients who were diagnosed with CAA based on imaging studies, including coronary angiography, echocardiography, CT, MRI and cardiac catheterisation, within 90 days (3 months) before or after the index date were enrolled.

In the case control study, cases included adult subjects who were hospitalised with a diagnosis of CAA between 1 January 2005 and 31 December 2011. Patients younger than 20 years and those who were diagnosed with CAA in the outpatient clinic were excluded. In the case group, the first hospitalisation date for CAA was assigned as the index date. For the control selection, 10 patients were randomly selected for each case by matching age, sex and index year of case diagnosis. The first admission date within the same year of CAA diagnosis was assigned as the index date for the control group.

### Survival status

According to the validation by Cheng et al of inhospital mortality in the NHIRD,[17] we defined death as '4' or 'A' status record on discharge and 'disenrollment' insurance status in the Registry for Beneficiaries. In contrast, patients admitted because of an emergency, but without follow-up data, were considered dead, which was also confirmed by checking the insurance status.

### Covariates

In the epidemiological study, we analysed age and sex, characteristics of subjects with respect to geographical region, urbanisation and income group at baseline. Income group was defined by the individual average monthly income during a period of 1 year before the index date and classified as low (< NTD\$20 000 or < US\$625) or high (≥ NT\$20 000 or ≥ US\$625). In the case control study, covariates for analysis included demographics and related risk factors. On the basis of previous evidence, related risk factors included coronary atherosclerosis,[8 18 19] hypertension,[18–21] dyslipidaemia,[19–21] diabetes mellitus,[19–22] cerebrovascular disease,[8] peripheral vascular disease,[8] varicose veins,[18 23] aortic dissection,[24] aortic aneurysm (AA),[24 25] systemic lupus erythematosus (SLE),[8] rheumatoid arthritis[8] and inflammatory bowel disease.[8] Furthermore, we defined atherosclerosis, hypertension and hyperlipidaemia as traditional cardiovascular risk factors.[20 21] A period of 1 year before the index date was reserved for covariate evaluation in both groups.

### Statistical analysis

Incidence rate, presented as number per $10^5$ person years, was defined as the number of patients with a newly diagnosed CAA, divided by the total Taiwanese population in each year. Mortality rate, presented as number per $10^5$ person years, was defined as the number of patients who died divided by the total Taiwanese population in each year. For trend analysis, the Cochran–Armitage test was used to detect any significant change in incidence rate or mortality rate during the study period. The demographic data of the total population in Taiwan were obtained from the Statistics, Department of Household Registration, Ministry of the Interior.[26] For descriptive statistics, continuous variables are presented as mean±SD; categorical variables are presented as frequency and proportion (%).

The OR and 95% CI for CAA were estimated by conditional logistic regression analysis for matched pair's data and were adjusted for the study covariates. The results were expressed as three models adjusted for different variables: model 1, demographics and traditional cardiovascular risk factors; model 2, demographics and related risk factors, excluding traditional cardiovascular risk factors; and model 3, demographics and related risk factors. Spearman's correlation coefficient was used to detect collinearity among variables. Data processing and statistical analysis were performed with SAS V.9.4 (SAS Inc, Cary, North Carolina, USA).

## RESULTS

### Epidemiological study population

From 2005 to 2011, there were 1493 patients who had at least two CAA diagnoses at an outpatient setting or one CAA diagnosis at an inpatient setting. After excluding 96 subjects who did not have any record of related examinations, a total of 1397 patients with a CAA diagnosis

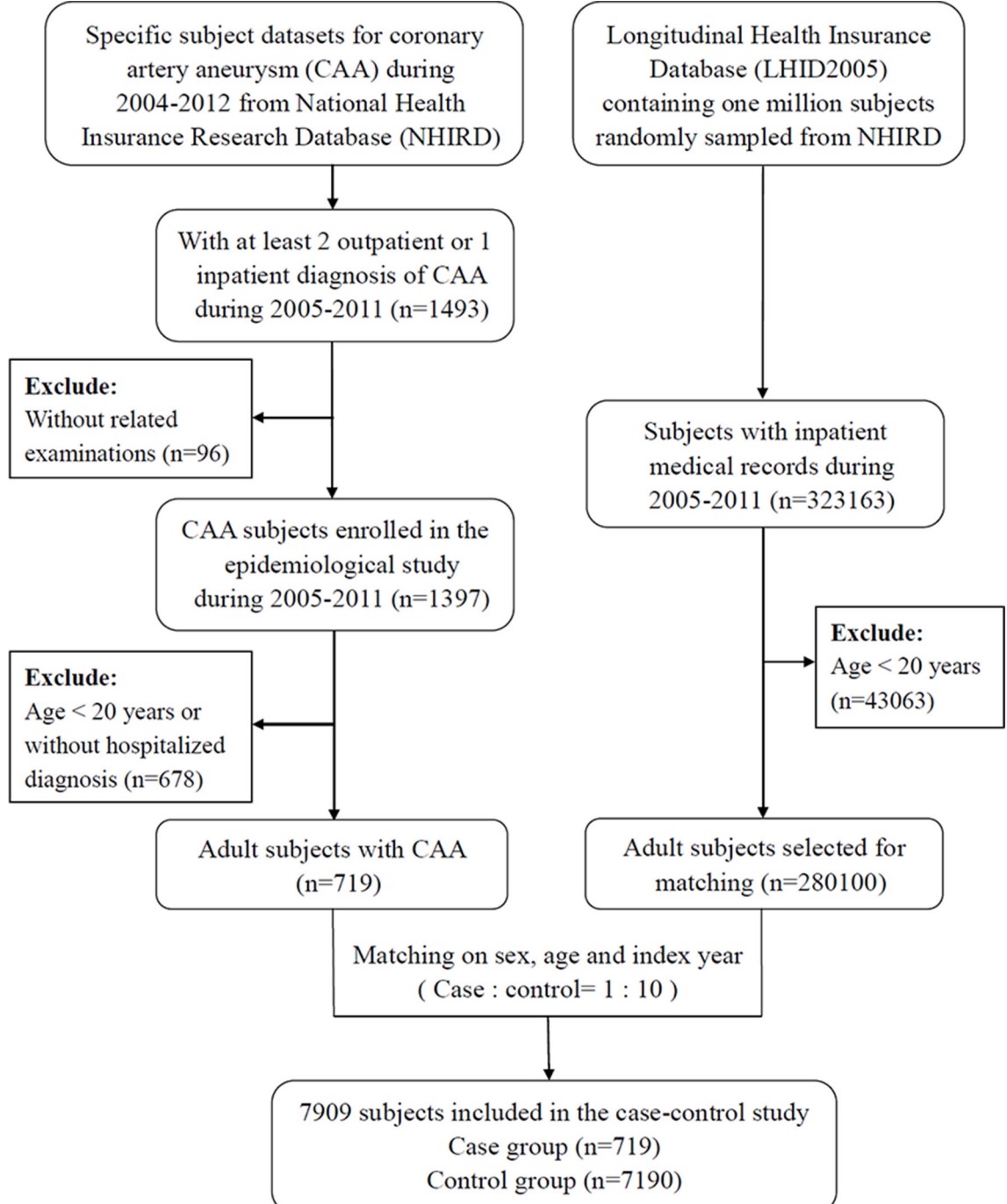

**Figure 1** Flow chart of the study population.

were enrolled in our study (Figure 1). Among the 1397 patients, mean age was 37.76±31.45 years. There were 586 paediatric patients (age <20 years), 430 adults (age ≥20 to <65 years) and 381 elderly patients (age ≥65 years), accounting for 41.9%, 30.8% and 27.3% of the CAA population, respectively (Table 1). Figure 2 shows the detailed age distribution where the majority of paediatric patients were <5 years of age; in particular, patients who were ≤1 year of age comprised 229 patients (23.2%). Furthermore, a male predilection (68.5%) was found, with a male to female ratio of 2.18. Most of the patients with CAA lived in northern Taiwan (39.1%), followed by

central (37.1%), southern (22.0%) and eastern (1.9%) areas; as high as 72.9% of the patients resided in urban areas. A majority (73.7%) of patients were in the low income group, which could be attributed to the predominance of young patients in this population.

### Incidence rate and mortality rate
The incidence rate of CAA from 2005 to 2011 in Taiwan was 0.87 per $10^5$ person years. About 200 patients were diagnosed with CAA every year. As shown in Figure 3, there was no significant difference in incidence rate from 2005 to 2011 in the trend test (p=0.2653).

**Table 1** Baseline characteristics of the coronary artery aneurysm population

| Variable (n=1397) | No (%) |
| --- | --- |
| Age (years) (mean±SD) | 37.76±31.45 |
| Age group (years) | |
| <20 | 586 (41.9) |
| 20–64 | 430 (30.8) |
| ≥65 | 381 (27.3) |
| Gender | |
| Men | 957 (68.5) |
| Women | 440 (31.5) |
| Area | |
| North | 546 (39.1) |
| Central | 518 (37.1) |
| South | 307 (22.0) |
| East | 26 (1.9) |
| Urbanisation | |
| Urbanised | 1019 (72.9) |
| Rural | 378 (27.1) |
| Income group* | |
| Low | 1030 (73.7) |
| High | 367 (26.3) |

*Income group was defined by the individual average monthly income during a 1 year period before the index date, and classified as low (<NTD\$20 000) and high (≥ NT\$20 000).

The mortality rate of the CAA population was 0.05 per $10^5$ person years. It increased slightly from 0.03 per $10^5$ person years in 2005 to 0.07 per $10^5$ person years in 2011 (p=0.0018; Figure 3). A total of 80 cases (17 adults and 63 elderly patients) died during the study period. No paediatric patient died; therefore, it may have attenuated the overall mortality. Specifically, the mortality rate in 811 adults was 9.9% (80 cases) at a mean follow-up period of 43.3±25.8 months.

### Case control study population

The selection process of subjects for the case and control groups is shown in Figure 1. Of 1397 patients identified in the previous epidemiological study, we enrolled 719 adult patients with a hospitalised diagnosis of CAA as the case group. After matching for age, sex and index year, there were 7190 subjects in the control group, yielding an overall population of 7909 subjects in the case control study.

Overall, the age and sex distributions were well balanced between the case and control groups (Table 2). The mean age of the case group was 62.57±13.91 years and 67.9% were men, which was similar to the control group. There was no significant difference in the urbanisation level and income group. Most of the patients lived in an urban area and were in the low income group.

### Risk factors associated with CAA

In conditional logistic regression analysis, the association between traditional cardiovascular risk factors and the presence of CAA was higher than in the general population (Table 2). The adjusted odds ratio (aOR) for coronary atherosclerosis (aOR 7.97; 95% CI 6.46 to 9.84), hypertension (aOR 2.09; 95% CI 1.73 to 2.53) and dyslipidaemia (aOR 2.48; 95% CI 2.06 to 2.99) were greater than twofold in model 1. By multivariate analysis in model 2, we found that diabetes mellitus (aOR 1.51; 95% CI 1.26 to 1.81), aortic dissection (aOR 6.76; 95% CI 1.89 to 24.14), AA (aOR 5.82; 95% CI: 2.02 to 16.83) and systemic lupus

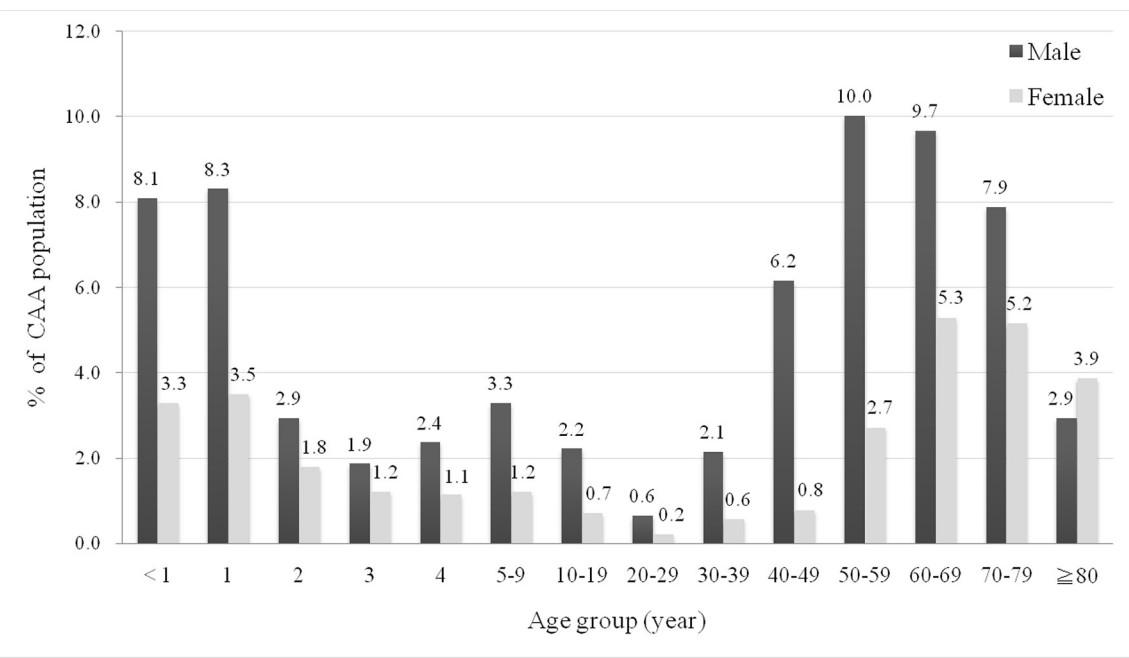

**Figure 2** Age distribution of the coronary artery aneurysm (CAA) population.

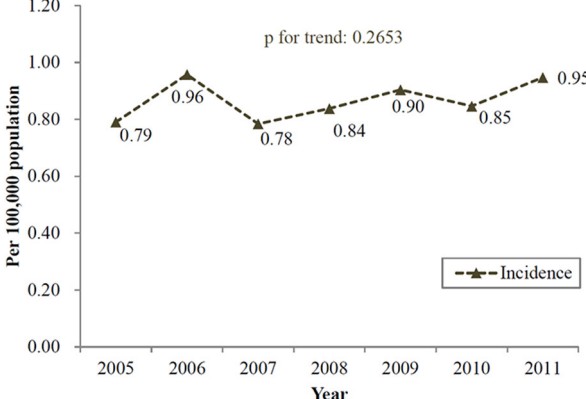

**(a)**

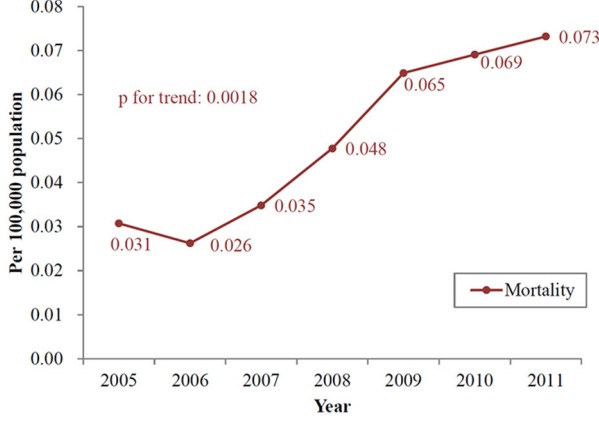

**(b)**

**Figure 3**  (A) Incidence rate of the coronary artery aneurysm (CAA) population during 2005–2011. (B) Mortality rate of the CAA population during 2005–2011.

erythematosus (aOR 4.09; 95% CI 1.32 to 12.62) were significantly associated with the presence of CAA. In model 3, traditional cardiovascular risk factors and AA were significantly associated with the presence of CAA. For the discrepancy, we found a moderate collinearity between diabetes mellitus and traditional cardiovascular risk factors (see online supplementary file 1).

Other vascular diseases, such as cerebrovascular disease, peripheral vascular disease and varicosities, were not statistically significant risk factors for CAA. Similarly, rheumatoid arthritis and inflammatory bowel disease failed to show significant differences between the two groups.

## DISCUSSION

The incidence rate of CAA from 2005 to 2011 in Taiwan was 0.87 per $10^5$ person years, with approximately 200 new cases every year. There was no increasing trend in the incidence rate of CAA during this period. In the USA, there were two studies on angiography, reporting a CAA prevalence of 0.2% (20/8422) between July 1975 and May 1979, and 4.9% (978/20087) from July 1981 to

February 1987.[1 2] A report from Greece for coronary angiography showed a CAA prevalence of 2.7% (287/10524) from 1995 to 2003.[3] Another case series conducted from 2011 to 2013 in Saudi Arabia found that 1115 patients with invasive coronary angiogram had 67 (6%) coronary artery ectasia cases.[4] Furthermore, in China, coronary artery ectasia were identified in 131 of 1400 older adults (prevalence 9.4%) when diagnosed by CT coronary angiography.[27] Two case series in Taiwan in 10120 and 924 patients accounted for a prevalence of 0.2% (25 cases) and 2.6% (24 cases), respectively.[5 6] Compared with other studies, with derived data from angiography case series, this study retrieved the CAA population from the NHIRD. Therefore, the lower incidence rate found in our study may be the fact that CAA is usually asymptomatic until the size becomes large enough to induce complications; only patients with ischaemic symptoms and admitted for further examinations will be diagnosed.

The mortality rate of the CAA population was 0.05 per $10^5$ person years, and all cases of death occurred in patients >20 years. For 811 adults, an overall mortality of 9.9% (80 cases) was noted in our study. The mortality of patients with CAA varies widely among different studies. The 5 year mortality rate reached as high as 29% based on a case series in the USA.[19] Similarly, the Coronary Artery Surgery Study[1] found that patients with CAA and a coexistent coronary stenosis had a 26% mortality rate at 5 years. In contrast, other countries had a lower mortality rate, ranging from 1.4% to 10.7%.[28–30] This discrepancy may be owing to a more severe obstructive coronary artery disease and longer follow-up time in the prior two studies. On the other hand, a small case series comprising 24 patients with CAA in Taiwan[6] reported a mortality rate of 9.1%, which was similar to the results of our study, despite a younger population and shorter follow-up time.

In contrast with previous angiography case series in adults, the present study had access to information on both paediatric and adult patients with CAA. The mean age of the paediatric population was 3.16±3.66 years. Up to 95.7% of children were diagnosed with Kawasaki disease. Similar to other epidemiological reports in Taiwan,[11 31 32] our results showed that most patients with Kawasaki disease were <5 years of age; infants ≤1 year of age comprised the majority of this population. Kawasaki disease is a systemic vasculitis that commonly involves the coronary arteries and is considered the main cause of non-atherosclerotic CAA in young children.[18 33]

In the present study, traditional cardiovascular risk factors were independently associated with CAA. Coronary atherosclerosis is a well established risk factor for CAA in the literature.[1 8 18] A small series containing 24 CAA patients in Taiwan reported over 60% patients with multi-vessel disease.[6] There are various aetiologies of CAA being extensively discussed, among which atherosclerosis remains the most common factor and accounts for 50% of acquired CAA in adults.[18 33] It has been hypothesised that atherosclerosis is linked to aneurysm formation through a process of inflammation, extending into the

**Table 2** OR for risk factors associated with coronary artery aneurysm (CAA)

| Variable | CAA (n=719) | Control (n=7190) | Crude OR | Model 1* Adjusted OR | Model 2[b] Adjusted OR | Model 3[c] Adjusted OR |
|---|---|---|---|---|---|---|
| Demographics† | | | | | | |
| Age | 62.57 (13.91) | 62.57 (13.92) | - | - | - | - |
| Male sex | 488 (67.9) | 4880 (67.9) | - | - | - | - |
| Urbanisation | | | | | | |
| Rural | 191 (26.6) | 1964 (27.3) | 1 | 1 | 1 | 1 |
| Urbanised | 528 (73.4) | 5226 (72.7) | 1.04 (0.87–1.24) | 0.95 (0.79–1.15) | 1.02 (0.85–1.22) | 0.95 (0.78–1.15) |
| Income group‡ | | | | | | |
| Low | 635 (88.3) | 6441 (89.6) | 1 | 1 | 1 | 1 |
| High | 84 (11.7) | 749 (10.4) | 1.15 (0.90–1.47) | 0.93 (0.70–1.23) | 1.14 (0.89–1.46) | 0.92 (0.69–1.21) |
| Risk factors | | | | | | |
| Coronary atherosclerosis | 244 (33.9) | 354 (4.9) | 10.82 (8.85–13.24)§ | 7.97 (6.46–9.84)§ | | 8.00 (6.47–9.90)§ |
| Hypertension | 459 (63.8) | 2836 (39.4) | 3.19 (2.68–3.80)§ | 2.09 (1.73–2.53)‡ | | 2.12 (1.75–2.58)§ |
| Dyslipidaemia | 289 (40.2) | 1126 (15.7) | 3.77 (3.19–4.45)§ | 2.48 (2.06–2.99)‡ | | 2.60 (2.15–3.14)§ |
| Diabetes mellitus | 188 (26.1) | 1457 (20.3) | 1.51 (1.26–1.80)§ | | 1.51 (1.26–1.81)§ | 0.82 (0.67–1.00) |
| Cerebrovascular disease | 74 (10.3) | 710 (9.9) | 1.05 (0.81–1.36) | | 1.02 (0.78–1.32) | 0.90 (0.68–1.20) |
| Peripheral vascular disease | 4 (0.6) | 59 (0.8) | 0.68 (0.25–1.87) | | 0.66 (0.24–1.82) | 0.47 (0.15–1.44) |
| Varicose vein | 2 (0.3) | 47 (0.7) | 0.43 (0.10–1.75) | | 0.46 (0.11–1.91) | 0.41 (0.09–1.90) |
| Aortic dissection | 4 (0.6) | 6 (0.1) | 6.67 (1.88–23.62)§ | | 6.76 (1.89–24.14)§ | 3.78 (0.77–18.49) |
| Aortic aneurysm | 6 (0.8) | 9 (0.1) | 6.67 (2.37–18.73)§ | | 5.82 (2.02–16.83)§ | 3.96 (1.19–13.20)§ |
| Systemic lupus erythematosus | 5 (0.7) | 10 (0.1) | 5.00 (1.71–14.63)§ | | 4.09 (1.32–12.62)§ | 3.26 (0.94–11.28) |
| Rheumatoid arthritis | 9 (1.3) | 48 (0.7) | 1.89 (0.92–3.86) | | 1.80 (0.87–3.73) | 1.40 (0.60–3.25) |
| Inflammatory bowel disease | 1 (0.1) | 31 (0.4) | 0.32 (0.04–2.36) | | 0.35 (0.05–2.54) | 0.49 (0.07–3.67) |

*Demographics and traditional cardiovascular risk factors; [b]demographics and related risk factors, excluding traditional cardiovascular risk factors; [c]all covariates.
†Demographics and traditional cardiovascular risk factors, excluding traditional cardiovascular risk factors.
†Age and gender for matching were not analysed in models. Different covariates were analysed in the three models:
‡Income group was defined by the individual average monthly income during a 1 year period before the index date, and classified as low (<NTD\$20 000) and high (≥ NT\$20 000).
§$p < 0.05$.

tunica media, which eventually causes degeneration of the cystic medial.[18]

Some studies have reported that hypertension was more common in patients with CAA,[20 34] whereas no significant difference was found in other series.[19 21] Only one study showed a lower frequency of hypertension in patients with CAA.[6] Nichols et al[18] indicated that hypertension was related to aortic and cerebral artery aneurysms; therefore, an association between hypertension and CAA might be expected. In addition, Markis et al[35] suggested that hypertension may play a role in the development of CAA, perhaps through accelerating the process of media destruction. Evidence indicates dyslipidaemia is a major contributor to atherosclerosis, which is characterised by the deposition of lipids and fibrous elements in the arteries and further formation of the atheroma.[36] In view of atherosclerosis, it may be reasonable that our study showed a positive association between dyslipidaemia and the presence of CAA.

It is noteworthy that an inverse association between diabetes and the presence of CAA had been reported in several studies.[19 20 22] A previous meta-analysis[37] indicated diabetes as a protective factor for the occurrence of CAA, with a pooled OR of 0.65 (95% CI 0.54 to 0.77). For a positive association, it was reported that hyperglycaemia can trigger arterial inflammation and increase the risk of atherosclerosis,[36] which is a risk factor for CAA. Nonetheless, a debate on whether CAA is a variant of atherosclerosis still exists.[38] Conversely, some authors proposed that the protective effect of diabetes may be from negative arterial remodelling[39] and increased matrix volume[37] of the coronary arteries. As a whole, the association between diabetes and CAA is currently not well defined.

In our study, the results showed that there were positive associations between aortic dissection, AA and SLE, and the presence of CAA. It had been speculated that CAA and AA share a similar histology and pathogenic process, which is generalised impairment of the wall of the entire arterial system.[40] The matrix metalloproteinases, a group of enzymes that degrade various components of the extracellular matrix in the arterial wall, have been found to be elevated both in patients with CAA and those with AA.[41 42] With respect to SLE, an increasing number of case reports have indicated its relationship with CAA in recent years.[43–46] In addition, SLE was found to be associated with AA.[47 48] It was suggested that CAA resulted from severe coronary inflammation through the deposition of immune globulin and complement.[49 50] SLE was thought to induce accelerated atherosclerosis, which is a risk factor for aneurysms.[51] Although there were reports suggesting a relationship between SLE and the development of CAA, the exact pathogenesis remains poorly understood.

To the best of our knowledge, this was the first population based study investigating the epidemiology and risk factors for CAA in Taiwan. Using a specific NHIRD dataset on all patients with CAA, we were able to provide an explicit age distribution, including both paediatric and adult populations. However, there were some limitations in the current study. First, we were unable to estimate the prevalence because not all patients in the epidemiological study were available for follow-up. Second, our data were obtained from a claim based database, which did not include information on aneurysm features, such as size, shape, location, number of aneurysms, degree of stenosis and lifestyle. In addition, we could not differentiate between aneurysm and ectasia in this study. Third, the control group selected from the general population may have had less opportunity to undergo examinations to detect atherosclerosis and AA. For this reason, the control group may have had fewer related events than the case group.

In conclusion, CAA patients were distributed in both the paediatric and adult populations in Taiwan. Kawasaki disease was predominant in children with CAA. Aside from the cardiovascular diseases, factors such as aortic disease and SLE may predispose adults to develop CAA. The results of this study are limited by the lack of available data on process and effectiveness of management, and medical therapy of CAA. Future studies are recommended to evaluate the pharmacoepidemiology of CAA.

**Acknowledgements** This study was based in part on data from the NHIRD provided by the Bureau of the NHI, Department of Health, and managed by National Health Research Institutes. The authors acknowledge the assistance of the Bureau of the NHI and National Health Research Institutes. This work was supported by the Ministry of Science and Technology (MOST 104–2320-B-037–035) and Kaohsiung Medical University Hospital (KMUH105-M512).

**Contributors** C-TF, Y-BH,and C-YC designed the data collection instruments, conceptualised and designed the study, drafted the initial manuscript and approved the final manuscript as submitted. C-CK, C-TF, Y-PF and CYC carried out the initial analyses and critically reviewed the manuscript. All authors approved the final manuscript as submitted.

**Funding** This work was supported by grants from the Ministry of Science and Technology(MOST 104-2320-B-037-035) and Kaohsiung Medical University Hospital (KMUH105-M512).

**Competing interests** None declared.

**Ethics approval** This study was approved by the institutional review board of Kaohsiung Medical University Hospital.

**Provenance and peer review** Not commissioned; externally peer reviewed.

**Data sharing statement** No additional data are available.

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
