## [Reviewer comments · BMJ Open]

ARTICLE DETAILS

TITLE (PROVISIONAL)	Epidemiology and risk factors of coronary artery aneurysm in Taiwan: A population-based case-control study
AUTHORS	Fang, Chein-Tang; Fang, Yi-Ping; Huang, Yaw-Bin; Kuo, Chen-Chun; CHEN, CHUNG-YU

VERSION 1 - REVIEW

REVIEWER	Yusuke Okubo a. Harvard TH Chan School of Public Health, Boston b. Department of Social Medicine, National Research Institute for Child Health and Development
REVIEW RETURNED	12-Jan-2017

GENERAL COMMENTS	[Abstract] # "Design: A retrospective case-cohort study" Is this really "case-cohort study"? I think this is typo. If so, please correct it. # Participants: "diagnostic code (ICD:414.11)" I believe this is ICD-9CM code. If so, please correct it. # "by logistic regression" It would be better to express "by conditional logistic regression". # "Otherwise, adjusted odds ratios" For what associations? # "Epidemiology of CAA in Taiwan was low" Please describe why the authors consider "the incidence of CAA in Taiwan was low". Did you compared with other countries? Or previous studies in Taiwan? [Introduction] # "Prevalence from 0.2% to 6.0" It should be "from 0.2% to 6.0%" # "Two other case series in Taiwan indicated that 0.25%-2.6% of patients had CAA." Please describe the baseline population of interest. # "At present, information regarding CAA and its risk factors is lacking in Asian countries." I don't agree with this statement because we know Kawasaki disease is a significant risk factor for the development of CAA, especially for children. Please revise this sentence.
--

[Methods]

"Current NHIRD and hospital regulations and guidelines did not mandate informed consent..."

This is due to anonymous nature of the database? If so, please describe it.

"covariate for analysis included demographics and related risk factors"

Please describe accurately how the authors identified the risk factors.

"Incidence, which was presented as number per 100,000, was defined..."

I think this should be "Incidence rate" since the denominator was the total Taiwanese population in each year.

"Categorical variables are presented as number and proportion (%)."

I think "number" should be changed to "frequency".

[Results]

"There were 586 pediatric patients"

In the method section, the author stated that "Patients younger than 20 years were excluded". It seems difficult to understand why pediatric patients were identified in the result section.

"a monthly insurance premium of less than 20,000 NTD"

Does this mean low income?

Please explain it since most readers except for Taiwanese do not know the value of NTD.

"Incidence and mortality"

They should be "Incidence rate and mortality rate"

"The mortality of the CAA population was low at..."

Please do not include judgmental comment "low" in the result section.

These judges should be described at discussion section.

"the age and sex distributions were well matched in the case and..."

well balanced between the case and....

"the association between traditional cardiovascular risk factors and..."

Please define "traditional cardiovascular risk factors in method section.

Multiple testing

The statistical tests in table 2 using conditional logistic regression have to be accounted for multiple testing, using Bonferroni correction or other methodology. This is because there were too many null hypotheses, and there would be possible false-positive results.

"in Model 3, a reverse association"

I cannot understand what this words mean.

the validity of conditional logistic regression analyses

I think the cases of peripheral vascular disease, varicose vein, aortic

	dissection/aneurysms, systemic lupus erythematosus, RA, and BID would be too small to include the regression model. It would be better to exclude these variables from the regression models. [Discussion] # "The average annual incidence of CAA" This is "annual incidence rate". # "Based on case series on angiography" When and where the studies were done? # "However, if we focused on the adult population, and overall mortality rate of 9.9% at...." - How did the authors calculate it? - Mortality rate should be AA per 100,000 person-years in its notation. # The author should understand the terms of epidemiology: mortality and mortality rate. - Mortality is cumulative incidence of death or risk of death - Mortality rate is incidence rate of death. The author's inconsistent description of "mortality" and "mortality rate" precludes readability for both clinician and epidemiologists. # Discussion section This section is too long and really redundant. Please revise them entirely. # Grammatical errors Numerous grammatical errors were observed in the draft. I recommend the authors use english-editing service to improve these errors and readability.
--	---

REVIEWER	Sunao Nakamura New Tokyo Hospital Japan
REVIEW RETURNED	06-Feb-2017

GENERAL COMMENTS	This paper reports coronary artery aneurysm covering is onset morphology to prognosis in a comprehensive manner, which is scientifically valuable. Its analysis of National Data is admirable. Considering such a finding is applied into treatment guideline or serves as a warning, author is requested to elucidate following points additionally. 1. In their prognosis study from 2005 to 2011, cause of mortality increase, particularly cause of mortality of adult cases. Was there any specific reason for the increase of mortality rate? 2. It is desirable to provide detail information about what was done after finding of CAA and its process as much as possible. Or may be many of them were in no need for treatment?
---

REVIEWER	Jay Pal University of Washington USA
REVIEW RETURNED	04-Apr-2017

GENERAL COMMENTS	This is an important manuscript investigating coronary artery aneurysms and related risk factors. There are some minor grammatical errors that should be corrected to improve readability.
--

VERSION 1 – AUTHOR RESPONSE

Reviewer 1

[Abstract]

1. The reviewer's comment: "Design: A retrospective case-cohort study" Is this really "case-cohort study"? I think this is typo. If so, please correct it.

1. The authors' answer: Thanks the reviewer's comment. The design is retrospective case-control study, we have corrected it in the manuscript. Please see abstract.

2. The reviewer's comment: Participants: "diagnostic code (ICD:414.11)"I believe this is ICD-9-CM code. If so, please correct it.

2. The authors' answer: Thanks the reviewer's suggestion. We used ICD-9-CM code: 414.11 to identify CAA patients. We have corrected it in the manuscript. Please see abstract.

3. The reviewer's comment: "by logistic regression" It would be better to express "by conditional logistic regression".

3. The authors' answer: Thanks the reviewer's comment. We have revised it in the manuscript. Please see abstract.

4. The reviewer's comment: "Otherwise, adjusted odds ratios" For what associations?

4. The authors' answer: Thanks the reviewer's comment. "Otherwise" is a redundant conjunction in the paragraph, we have deleted it in the manuscript. Please see abstract.

5. The reviewer's comment: "Epidemiology of CAA in Taiwan was low" Please describe why the authors consider "the incidence of CAA in Taiwan was low". Did you compare with other countries? Or previous studies in Taiwan?

5. The authors' answer: Thanks the reviewer's comment. In the introduction and discussion section, we reviewed several angiographic series and a prevalence from 0.2% to 6.0% was yielded among different countries. However, it is hard to compare our epidemiology data with other countries due to limited information from other countries. Therefore, for decreasing confusedly in reading our study, we have modified our text in revised manuscript as "In Taiwan, CAA patients were distributed both in pediatric and adult population." Please see abstract.

[Introduction]

6. The reviewer's comment: "Prevalence from 0.2% to 6.0" It should be "from 0.2% to 6.0%"

6. The authors' answer: Thanks the reviewer's suggestion. We have revised it in the manuscript. Please see page 4 L6.

7. The reviewer's comment: "Two other case series in Taiwan indicated that 0.25%-2.6% of patients had CAA." Please describe the baseline population of interest.

7. The authors' answer: Thanks the reviewer's comment. We had cited the references in the content. An angiographic series by Wang et al. reported 25 patients (0.25%, 25/10120) had CAA. The mean age was 64 years and 84% were male. Eighty percent of the patients presented atypical angina or

acute coronary syndrome. Most patients had hypertension (60%), followed by cigarette smoking (52%), dyslipidemia (28%) and diabetes (8%). The other series by Yip et al. enrolled 924 AMI patients undergoing PCI, and 24 patients (2.6%) had aneurysmal dilatation. The mean age was 53±13 with 95.8% male sex. The concomitant diseases were hypercholesterolemia (62.5%), hypertension (41.7%) and diabetes (12.5%). Please see Page 4 L6-9.

8. The reviewer's comment: "At present, information regarding CAA and its risk factors is lacking in Asian countries. "I don't agree with this statement because we know Kawasaki disease is a significant risk factor for the development of CAA, especially for children. Please revise this sentence.

8. The authors' answer: Thanks the reviewer's comment. As the reviewer said, Kawasaki disease was a known risk factor for the development of CAA. For better readability, we would revise the description as "Despite Kawasaki disease is a significant risk factor for the development of CAA, especially for children, information regarding CAA and its related risk factors in adult population is limited in Asian countries. ". Please see Page 4 L18-20.

[Methods]

9. The reviewer's comment: "Current NHIRD and hospital regulations and guidelines did not mandate informed consent..."This is due to anonymous nature of the database? If so, please describe it.

9. The authors' answer: Thanks the reviewer's comment. Data in the NHIRD that could be used to identify patients or care providers, including medical institutions and physicians, is scrambled before being sent to the National Health Research Institutes for database construction and is further scrambled before being released to each researcher. Theoretically, it is impossible to query the data alone to identify individuals at any level using this database. All researchers who wish to use the NHIRD and its data subsets are required to sign a written agreement declaring that they have no intention of attempting to obtain information that could potentially violate the privacy of patients or care providers. For above reason, we would revise the description as "NHIRD data were de-identified by scrambling the identification codes of both patients and medical facilities, and released to the public for research purposes. Therefore, current NHIRD and hospital regulations and guidelines did not mandate informed consent in this retrospective case-control study due to we used anonymous nature of the database." . Please see Page 5 L12-16.

10. The reviewer's comment: "covariate for analysis included demographics and related risk factors" Please describe accurately how the authors identified the risk factors.

10. The authors' answer: Thanks the reviewer's comment. We identified risk factors through literature review and cited the references in the content. Two review articles mentioned atherosclerosis, inflammatory disorder and connective tissue disorder were related to CAA based on its etiology. Hypertension, hyperlipidemia, diabetes and smoking were noted in other studies. In addition, a study found varicose veins more common in patients with coronary artery ectasia. In the current study, we define atherosclerosis, hypertension, hyperlipidemia and diabetes as traditional cardiovascular risk factors. Therefore, for decreasing confusedly in reading our study, we have modified our text in revised manuscript as "In the case-control study, covariates for analysis included demographics and related risk factors. On the basis of previous evidences, related risk factors included coronary atherosclerosis,^{8, 18, 19} hypertension,¹⁸⁻²¹ dyslipidemia,¹⁹⁻²¹ diabetes mellitus,¹⁹⁻²² cerebrovascular disease,⁸ peripheral vascular disease,⁸ varicose vein,^{18, 23} aortic dissection,²⁴ aortic aneurysm (AA),^{24, 25} systemic lupus erythematosus (SLE),⁸ rheumatoid arthritis,⁸ inflammatory bowel disease.⁸" Please see Page 7 L5-10.

11. The reviewer's comment: "Incidence, which was presented as number per 100,000, was defined..."I think this should be "Incidence rate" since the denominator was the total Taiwanese population in each year.

11. The authors' answer: Thanks the reviewer's suggestion. We have modified incidence to incidence rate in the revised manuscript.

12. The reviewer's comment: "Categorical variables are presented as number and proportion (%)." I think "number" should be changed to "frequency".

12. The authors' answer: Thanks the reviewer's suggestion. We have revised it in the manuscript. Please see Page 8 L4.

[Results]

13. The reviewer's comment: "There were 586 pediatric patients" In the method section, the author stated that "Patients younger than 20 years were excluded". It seems difficult to understand why pediatric patients were identified in the result section.

13. The authors' answer: Thanks the reviewer's comment. The current study was separated into two parts: epidemiological and case-control study. First, we enrolled 1397 patients, including pediatric and adult population, in the epidemiological study. Then, we only kept 811 adult patients in the case-control study to assess risk factors.

14. The reviewer's comment: "a monthly insurance premium of less than 20,000 NTD" Does this mean low income? Please explain it since most readers except for Taiwanese do not know the value of NTD.

14. The authors' answer: Thanks the reviewer's comment. We have described the definition of income group in detail in method and result section, table 1 and table 2 in revised manuscript. Please see Page 7 L3-5 and Page 9 L7-9.

15. The reviewer's comment: "Incidence and mortality" They should be "Incidence rate and mortality rate"

15. The authors' answer: Thanks the reviewer's suggestion. We have revised it in the manuscript.

16. The reviewer's comment: "The mortality of the CAA population was low at..." Please do not include judgmental comment "low" in the result section. These judges should be described at discussion section.

16. The authors' answer: Thanks the reviewer's suggestion. We have revised it in the manuscript.

17. The reviewer's comment: "the age and sex distributions were well matched in the case and..." well balanced between the case and....

17. The authors' answer: Thanks the reviewer's suggestion. We have revised it in the manuscript.

18. The reviewer's comment: "the association between traditional cardiovascular risk factors and..." Please define "traditional cardiovascular risk factors in method section.

18. The authors' answer: Thanks the reviewer's comment. We define atherosclerosis, hypertension, hyperlipidemia as traditional cardiovascular risk factors in the current study. We have added the definition in method section. Please see page 7 L7-12.

19. The reviewer's comment: Multiple testing: The statistical tests in table 2 using conditional logistic regression have to be accounted for multiple testing, using Bonferroni correction or other methodology. This is because there were too many null hypotheses, and there would be possible false-positive results.

19. The authors' answer: Thanks the reviewer's comment. In fact, we had test multiple testing by using Bonferroni correction and different methods. However, there is no significantly difference between using multiple testing and no using multiple testing to discuss our results.

20. The reviewer's comment: "in Model 3, a reverse association" I cannot understand what this words mean.

20. The authors' answer: Thanks the reviewer's comment. For decreasing confusedly in reading our study, we have modified our text in revised manuscript as "In Model 3, traditional cardiovascular risk factors and AA were significantly associated with the presence of CAA." Please see Page 13 L2-3.

21. The reviewer's comment: the validity of conditional logistic regression analyses. I think the cases of peripheral vascular disease, varicose vein, aortic dissection/aneurysms, systemic lupus erythematosus, RA, and BID would be too small to include the regression model. It would be better to exclude these variables from the regression models.

21. The authors' answer: Thanks the reviewer's excellent comment. We very agree reviewer's comment that case number of factors (peripheral vascular disease, varicose vein, aortic dissection/aneurysms, systemic lupus erythematosus, RA, and BID) are too small in our study and regression model. In fact, due to small case numbers in regression which cause some bias, we divided our model into 3 to test our hypothesis. As reviewer's suggestion, model 1 excluded these small sample size factors to test our outcome.

[Discussion]

22. The reviewer's comment: "The average annual incidence of CAA" This is "annual incidence rate".

22. The authors' answer: Thanks the reviewer's suggestion. We have revised it in the manuscript.

23. The reviewer's comment: "Based on case series on angiography" When and where the studies were done?

23. The authors' answer: Thanks the reviewer's comment. For these case series, we have cited their references in the sentence. In America, there were two series reporting CAA prevalence of 0.2% and 4.9% in 1990 and 1983, respectively. More recently, a report performed in Greece showed a prevalence of 2.7% in 2006. In 2015, another series conducted in Saudi Arabia found 6% of the patients had coronary artery ectasia and it was higher than other published studies. We have revised it in the manuscript. Please see page 14 L4-11.

24. The reviewer's comment: "However, if we focused on the adult population, and overall mortality rate of 9.9% at...."

– How did the authors calculate it?

– Mortality rate should be AA per 100,000 person-years in its notation.

24. The authors' answer: Thanks the reviewer's comment. We have revised the description as "mortality" instead of "mortality rate" in the revised manuscript. The overall mortality was calculated below: adult death cases among 2005-2011 divided by the adult population (80/811). Please see page 9 L18-19 and page 14 L19.

25. The reviewer's comment: The author should understand the terms of epidemiology: mortality and mortality rate.

– Mortality is cumulative incidence of death or risk of death

– Mortality rate is incidence rate of death.

The author's inconsistent description of "mortality" and "mortality rate" precludes readability for both clinician and epidemiologists.

25. The authors' answer: Thanks the reviewer's comment. We have made it clear and revised it in the manuscript.

26. The reviewer's comment: Discussion section: This section is too long and really redundant. Please revise them entirely.

26. The authors' answer: Thanks the reviewer's comment. We have shorted discussion section in the revised manuscript.

27. The reviewer's comment: Grammatical errors: Numerous grammatical errors were observed in the

draft. I recommend the authors use English-editing service to improve these errors and readability.
 27. The authors' answer: Thanks the reviewer's comment. The entire manuscript has been corrected by a native English language expert.

Reviewer 2

1. The reviewer's comment: In their prognosis study from 2005 to 2011, cause of mortality increase, particularly cause of mortality of adult cases. Was there any specific reason for the increase of mortality rate?

1. The authors' answer: Thanks the reviewer's comment. Mortality rate slightly increased from 2005 to 2011, which may cause by cases receiving different management or medical therapy. However, we lack available data on management and medical therapy of study population. It is hard for us evaluating the reason of increasing mortality rate in our study. Furthermore, we only can evaluate all caused mortality rate from NHIRD, it is hard to explain the specific disease or CAA caused death in our population. Otherwise, we speculated that mortality rate slightly increased which cause might be disease progression as time went by. In hence, from our database and small number of deaths in our study, it is hard to infer any specific or correct reason for the increase of mortality rate.

2. The reviewer's comment: It is desirable to provide detail information about what was done after finding of CAA and its process as much as possible. Or may be many of them were in no need for treatment?

2. The authors' answer: Thanks the reviewer's comment. In fact, we think reviewer's comment is very important for our research. However, the results of this study are limited by the lack of available data on process and effectiveness of management and medical therapy of CAA. Future studies are recommended to evaluate pharmacoepidemiology of CAA. Please see page 18 L17-19.

Reviewer 3

1. The reviewer's comment: There are some minor grammatical errors that should be corrected to improve readability.

1. The authors' answer: Thanks the reviewer's comment. The entire manuscript has been corrected by a native English language expert.

We acknowledge the reviewer's comments and suggestions very much, which are valuable in improving the quality of our manuscript.

VERSION 2 – REVIEW

REVIEWER	Yusuke Okubo National Research Institute of Child Health and Development, Japan Harvard T.H.Chan School of Public Health, USA
REVIEW RETURNED	04-May-2017

GENERAL COMMENTS	The autohrs dealt with the reviewer's comments appropriately, and there is no specific comment for the revision.
--

REVIEWER	Sunao Nakamura New Tokyo Hospital, Japan
REVIEW RETURNED	18-May-2017

GENERAL COMMENTS	This paper reports coronary artery aneurysm covering is onset
---

	morphology to prognosis in a comprehensive manner, which is scientifically valuable. Its analysis of National Data is admirable. This paper is appropriate for this journal.
--	--